

# Query sampler: generating query sets for analyzing search engines using keyword research tools

Sebastian Schultheiß[1], Dirk Lewandowski[1,2], Sonja von Mach[1] and Nurce Yagci[1]

[1] Department of Information, Hamburg University of Applied Sciences, Hamburg, Germany
[2] Department of Computer Science and Applied Cognitive Science, University Duisburg-Essen, Duisburg, Germany

## ABSTRACT

Search engine queries are the starting point for studies in different fields, such as health or political science. These studies usually aim to make statements about social phenomena. However, the queries used in the studies are often created rather unsystematically and do not correspond to actual user behavior. Therefore, the evidential value of the studies must be questioned. We address this problem by developing an approach (query sampler) to sample queries from commercial search engines, using keyword research tools designed to support search engine marketing. This allows us to generate large numbers of queries related to a given topic and derive information on how often each keyword is searched for, that is, the query volume. We empirically test our approach with queries from two published studies, and the results show that the number of queries and total search volume could be considerably expanded. Our approach has a wide range of applications for studies that seek to draw conclusions about social phenomena using search engine queries. The approach can be applied flexibly to different topics and is relatively straightforward to implement, as we provide the code for querying Google Ads API. Limitations are that the approach needs to be tested with a broader range of topics and thoroughly checked for problems with topic drift and the role of close variants provided by keyword research tools.

## INTRODUCTION

When investigating social phenomena in search engines, researchers build lists of queries, which they enter in the search engine and analyze the returned results. For instance, to determine whether Google prefers results of a particular political leaning, a list of queries can be designed, and the returned results analyzed. However, a central issue is whether the study results would be valid if another set of queries on the same topic was used. In the literature, we find that query sets are developed rather unsystematically, and therefore, the evidential value of such studies may be questioned. We address this research gap by developing an approach to sample queries from commercial search engines, including information on how often each query is entered, using commercial keyword research tools provided by search engine companies, such as Google.

Corresponding author
Sebastian Schultheiß,
sebastian.schultheiss@haw-hamburg.de

A myriad of studies uses query lists to generate search results to be further analyzed. In the information retrieval and information science communities, this is a standard procedure (*e.g.*, *Verma & Yilmaz, 2016*; *Fu, 2017*). However, such query lists are also used in other fields. For instance, in medicine, studies examine the information quality of search results, for example, related to cancer diet (*Herth et al., 2016*) or breast cancer (*Janssen et al., 2018*). Moreover, in media and communications, researchers may aim to identify fake news for COVID-19-related queries (*Mazzeo, Rapisarda & Giuffrida, 2021*). An example from computational social sciences is a study investigating vaccine-related webpages using predefined terms such as "vaccine + danger" (*Xu, 2019*). Other exemplary studies assess Wikipedia's coverage when searching for psychological concepts (*Schweitzer, 2008*), examine results of political queries during election campaigns (*Unkel & Haim, 2021*), and in the field of sociology, analyze public information-seeking related to gun control and gun rights (*Semenza & Bernau, 2022*). The common aspects of these studies are that authors use lists of search terms, enter the terms into Google or other search engines, and analyze the retrieved results according to their research questions. The question is where these queries come from. How do researchers generate query sets that reflect what users search for? To what degree do researchers succeed in assembling such query sets? In some studies, the authors identify this problem and discuss whether the queries used in their research, such as medical or political terms, may not fully match the phrases people actually use when searching for information on the respective topic (*Herth et al., 2016*; *Unkel & Haim, 2021*; *Semenza & Bernau, 2022*). Inevitably, the question arises: Would the respective study have yielded different results if the authors had used other queries?

From the thematic diversity of studies using query sets and the problem of considering real search queries, two requirements of an ideal query set are derived: (1) consideration of query popularity and (2) topic coverage. First, the query set should reflect the users' search interests by considering the search volume (popularity) of the queries. This prerequisite is essential for studies that aim to draw conclusions about social phenomena, for example, from health (*Janssen et al., 2018*) or political science (*Unkel & Haim, 2021*). Second, the ideal query set should cover the queries on any topic as comprehensively as possible.

Queries submitted by real search engine users of all search engines worldwide can be considered the "ideal" of a query set. These queries would be most likely yield valid information about social phenomena since they would map the search interests of the entire online society and would be independent of specific search engine providers. The closest to this holy grail would be the use of panel data from permanently observed users. Such data, however, would be only a sample and no longer a complete survey. While such a sample may be appropriate for many studies, it may not be sufficient when the aim is to cover topics in their entirety, especially with long-tail phenomena. Another promising approach is the Archive Query Log (AQL). Once the data collection is completed, AQL will contain approximately 356 million queries from several search engines extracted from the Internet Archive (*Reimer et al., 2023*). Instead of data from multiple search engines, some researchers have access to queries from individual search engine providers. However, such data are not freely available to researchers, and studies based on this data usually focus on optimizing the respective search systems (*e.g.*, *Dang, Kumaran & Troy, 2012*; *Baytin et*

*al., 2013*). Thus, a central methodological challenge is to find alternative approaches for developing query sets.

## Research questions

In this article, we answer the following research questions (RQs):

**RQ1**: What approaches are used by scholars to generate query sets, and what are the strengths and weaknesses of these approaches?

**RQ2**: What is a suitable approach for generating a query set that considers query popularity and allows the coverage of a specific topic?

## Research gap

Scholars reuse or generate query sets by applying numerous methods. In information retrieval research, test collections consisting of queries, documents, and relevance judgments are essential (for overview of Text Retrieval Conference (TREC) test collections see *Harman & Voorhees, 2007*). However, queries in test collections are static, limited to specific languages, and partially prefiltered, even if a test collection includes queries provided by a search engine provider (*e.g.*, MS MARCO dataset with Bing data; *Craswell et al., 2021*).

We found seven types of query sets researchers generate:

1. Queries delivered by a search engine provider, for example, Microsoft (*Azzopardi et al., 2020*).
2. Popularity data; this includes the use of all tools that provide data on query popularity, while the scope of functions differs greatly in these tools, for example, *Alby (2020)*.
3. Autocomplete suggestions users receive as a drop-down list when entering queries, for example, *Haak & Schaer (2022)*.
4. Content extracted from online communities, such as the AskDocs section of Reddit (*Zuccon et al., 2016*).
5. Queries provided by subjects asked to generate queries, for example, utilizing an online survey (*Bilal & Ellis, 2011*).
6. Queries developed by the study authors based on specific criteria, for example, query type (*Schultheiß, Sünkler & Lewandowski, 2018*).
7. Predetermined lists of terms, for example, names of political candidates (*Hinz, Sünkler & Lewandowski, 2023*).

The commonality in the mentioned approaches is that they are only partly suitable for drawing conclusions about social phenomena since they were either created for other objectives (*e.g.*, test collections for information retrieval research, as described above), contain terms that users may not use (*e.g.*, lists of technical jargon), or are limited in scalability (*e.g.*, content from online communities). The overall problem with these approaches is that the query sets represent actual user querying behavior in some way, but researchers cannot guarantee that they represent the entire user population (*e.g.*, all users or a particular user group). Therefore, the evidential value of such studies can be questioned.

In this study, we address this issue and present an approach that allows building extensive sets of queries on any topic. We do this by using lists of terms and keyword research

tools. Such tools are offered by search engine marketing companies (*e.g.*, Semrush) and commercial search engine companies, such as Google, to allow marketers to plan their campaigns (see 'Selecting a keyword research tool'). Keyword research tools find additional keywords based on already known relevant keywords; that is, the tools make suggestions for more keywords that can be used to address customers. The tools also predict the query volume for each search query, that is, the predicted number of searches per month. By offering keyword ideas, the tools aim to support marketers. In this context, keywords are terms or phrases a website should show up on a search engine, while queries refer to the actions of the users (https://www.searchenginejournal.com/understanding-difference-queries-keywords/126421/#close). In this article, we will use both terms synonymously.

### Structure of the article

The rest of the article is structured as follows. First, we show which approaches for creating a search query set have been used in the literature and their advantages and disadvantages. Subsequently, we describe our approach to generating a query set. The approach consists of (A) selecting an initial list of terms, (B) including synonyms and alternative spellings, (C) selecting a keyword research tool, and (D) generating keyword ideas. The description of the approach is followed by its empirical verification using query data from two published studies where queries formed the basis of analysis (*Herth et al., 2016*; *Lewandowski, Sünkler & Yagci, 2021*). Finally, we discuss the results and present suggestions for future research.

## LITERATURE REVIEW

Through an extensive literature search, we identified seven approaches to generating query sets. Using Scopus and Google Scholar, we searched for articles containing words related to query sets (*e.g.*, set of queries, list of queries) together with words related to search engines (*e.g.*, search engine, Google). We focused on finding articles describing how the query sets were built, regardless of the study's objective. Table 1 details these approaches, summarizing the extent to which the prerequisites of popularity consideration and topic coverage are considered. The following sections describe the approaches we identified in the literature in more detail.

### Queries delivered by a search engine provider

Using queries delivered by search engine providers is the most promising of the feasible approaches listed in Table 1 since both criteria (popularity consideration and topic coverage) are met. Examples are studies where researchers have access to transaction logs from search engines such as Google (*Kinney, Huffman & Zhai, 2008*), Bing (*Dang, Kumaran & Troy, 2012*; *Das et al., 2017*), Yahoo (*Goel et al., 2010*), AOL (*Lucchese et al., 2013*), Yandex (*Baytin et al., 2013*), Excite (*Gravano, Hatzivassiloglou & Lichtenstein, 2003*), Sogou (*Whiting, Jose & Alonso, 2016*), or T-Online (*Lewandowski, 2015*). The authors of these studies focus on improving the performance of their company's search systems, for example, regarding autocorrection (*Baytin et al., 2013*) or query reformulations (*Dang, Kumaran & Troy, 2012*). A significant disadvantage is that analyses depend on search engine providers' willingness to provide researchers

**Table 1  Approaches for generating a query set.**

| Approach | Criteria | | Maximum number of queries[a] |
|---|---|---|---|
| | **Popularity consideration** | **Topic coverage** | |
| Queries of all search engines worldwide[b] | Popularity of topics is considered *across search engines* | All conceivable topics are considered, *independent* of specific search engines | / |
| Queries delivered by a search engine provider | Popularity of topics is considered within a specific search engine[c] | All conceivable topics searched for in the respective search engine are considered | 2,6 B (*Goel et al., 2010*) |
| *Query sampler, combining the approaches "popularity data" and "predetermined lists of terms" (our approach)* | *Popularity is considered through data on search volume using the Google Ads API* | *Most topics are covered (see 'Selecting a keyword research tool' for information on restrictions regarding search volume data for specific topics)* | *Potentially unlimited* |
| Popularity data | Popularity is considered through data on search volume | Most topics are covered (see 'Selecting a keyword research tool' for information on restrictions regarding search volume data for specific topics) | 29,132 (*D'Ambrosio et al., 2015*) |
| Autocomplete suggestions | Popularity is considered, but no data on search volume are given | Most topics are covered (see 'Autocomplete suggestions' for information on restrictions regarding autocompletion for specific topics) | 21,407 (*Haak & Schaer, 2022*) |
| Content extracted from online communities | Popularity is only reflected within the respective online community | Only topics that are discussed in the online community are covered | 10,717 (*Yilmaz et al., 2019*) |
| Queries provided by subjects | The queries are not created in a natural environment (*e.g.*, within an online forum or by crowdsourcing) | Theoretically, an unlimited coverage can be achieved | 5,764 (*Bailey et al., 2016*) |
| Queries developed by the study authors | Popularity is not considered | Queries are mostly arbitrarily arranged without specific topics being covered in depth | 50 (*McCreadie et al., 2012*) |
| Predetermined lists of terms | Depends on the list (see 'Predetermined lists of terms') | Depends on the list (see 'Predetermined lists of terms') | 6,211 (*Hinz, Sünkler & Lewandowski, 2023*) |

**Notes.**
[a]The sizes only refer to those in cited studies.
[b]This approach was not identified in the literature, but it represents the best imaginable, albeit unrealistic, fulfilment of both criteria.
[c]This only applies if the data come from a popular search engine (*e.g.*, Google or Bing), assuming that data from this search engine are representative of general search behavior.

with data. It is highly unlikely that search engine providers grant access to their data when researchers wish to investigate topics that are outside the providers' self-interests (*Lewandowski, Sünkler & Schultheiß, S, 2020*). Thus, independent decisions by researchers regarding the thematic and quantitative scope of the data are barely possible. Furthermore, the provided queries refer only to a single search engine without allowing comparisons between different search engines. Even when researchers have access to transaction logs of multiple search engines, comparing the data is quite challenging. *Jansen & Spink (2006)* enumerated differences between nine transaction logs. These differences concern, for

example, different time spans when the logs were created and missing numbers of sessions and terms in two logs.

## Popularity data

Since most researchers do not have access to queries from search engine providers, one possible solution is popularity data. Popularity data contain queries including information on their popularity (*i.e.,* search volume), with the accuracy of search volume data varying considerably.

Popularity data are made available through tools, mainly Google Trends. Other studies use keyword research tools, such as Google Keyword Planner. Predetermined terms or lists of terms, for example, on specific topics, serve as the basis (*i.e.,* seed terms, see 'Selecting a keyword research tool'). This differentiates the popularity data approach from the predetermined lists approach (see 'Predetermined lists of terms'), in which the list entries are synonymous with the queries used. The studies by *Ballatore (2015)* and *Fumagalli, Bailoni & Giunchiglia (2020)* are examples of studies using Google Trends. *Ballatore (2015)* selected the most popular queries from Google Trends for several conspiracy theories, while *Fumagalli, Bailoni & Giunchiglia (2020)* used Google Trends to generate queries relating to Schema.org types, for example, book series or creative work. *Tana (2018)* used Google Trends to retrieve the top queries for seed terms such as "depression." Google Trends allows access to actual searched terms during a specific time episode. However, Google Trends provides not absolute but normalized data that express the search volume of the respective term in relation to the search volume of all other searches at a given time (https://support.google.com/trends/answer/4365533?hl=en). Absolute numbers on search volume are provided by keyword research tools such as Google Keyword Planner (https://support.google.com/google-ads/answer/7337243?hl=en). For instance, in a study commissioned by a German health insurer (*Central, 2015*), the authors used a predetermined list of $N = 50$ common diseases as seed terms for forming term clusters consisting of the disease terms (*e.g.,* hyperkinetic disorder) as well as frequently used synonyms (*e.g.,* ADHD) and additional terms (*e.g.,* doctor) by using a keyword research tool.[1] Similar approaches were taken by *Alby (2020)* and *D'Ambrosio et al. (2015)*. Regarding the topic of skin diseases, *Alby (2020)* used disease terms and synonyms to build search queries ($N = 2,397$) *via* Google Keyword Planner, while *D'Ambrosio et al. (2015)* used preconception-related keywords to obtain queries ($N = 29,132$) that were actually searched for by Italian Internet users. For Google Keyword Planner, however, the limitation must be added that Google greatly reduced the accuracy of the data for accounts with low AdWords sales in 2016. Since then, search volumes are provided only in broad ranges (*e.g.,* 10–100 or 100–1K average monthly searches) (https://www.seroundtable.com/google-keyword-planner-throttled-22535.html). This means that studies similar to the pre-2016 studies using Google Keyword Planner could not be conducted anymore, at least not with accurate search volume data. *Waller (2011)* used data from web analytics company Hitwise (now a division of Connexity).[2] The sample covered queries typed into Google Australia over a 4-week period in 2009.

[1] The authors did not mention Google Keyword Planner explicitly, but its usage can be assumed.

[2] Connexity: https://connexity.com/ (Hitwise was acquired by Connexity in 2015.)

Exact values for the search volume of individual keywords are also provided by Keyword Magic Tool (https://de.semrush.com/analytics/keywordmagic/?q=adhs{&}db=de) from Semrush. Some tools use more than one source to provide their keywords, but the exact methods from which sources keywords and search volume are generated are kept secret. However, not having insights into the origin of the data is a serious issue in academic research.

## Autocomplete suggestions

Search engines deliver ideas to help the user to formulate their information need. Users receive common queries as a drop-down *via* autocomplete suggestions. The predictions match what a user started to enter and incorporate other factors, such as trending interest in the query. Search engines do not provide autocomplete suggestions for all content. For example, Google prevents predictions that are in violation of Google policies, such as sexually explicit, dangerous, or harassing content (https://support.google.com/websearch/answer/7368877). The same holds for Bing, as it filters spam and adult and offensive content from the suggestions (https://blogs.bing.com/search/2013/03/25/a-deeper-look-at-autosuggest). Nevertheless, in some cases, autocomplete suggestions can contain misinformation that may hurt organizations or individuals (*Hiemstra, 2020*).

Autocomplete predictions are used by researchers to create query sets. *Haak & Schaer (2022)* crawled $N = 21,407$ autocomplete suggestions from Google to analyze person-related suggestions for biases. *Wu et al. (2016)* developed a system that discovers query patterns (*e.g.*, "jobs in [location]") by using query autocomplete features. The authors aimed to discover a focused set of queries that center around an entity. In contrast, *Bar-Yossef & Gurevich (2008)* developed algorithms for sampling random autocomplete suggestions. *Fumagalli, Bailoni & Giunchiglia (2020)* used "Answer the public" (https://answerthepublic.com/), a tool that uses autocomplete suggestions from Google and Bing and organizes the queries according to different criteria, such as question type. *Haider (2016)* used autocomplete to define the queries for her study on informational structures on waste sorting.

Since autocomplete suggestions rely on actions taken by users, real user behavior is reflected. However, since the suggestions come from the search engine provider, the underlying algorithm and ranking factors for the autocomplete predictions can only be understood rudimentarily from the outside. In addition, it is crucial to remember that creating suggestions is complex, based on many influencing factors, such as a user's past searches. Furthermore, suggestions excluded due to inappropriate content, as described above, result in incomplete sets of user searches.

## Content extracted from online communities

Using content from online communities differs significantly from the previous approaches, as no queries from web search engines are considered. Instead, based on discussions in online communities, researchers map popular search queries for the topics discussed.

*Yilmaz et al. (2019)* used questions posted on an educational Q&A website to build a query set in the Turkish language. Related to medical topics, *Zuccon et al. (2016)* created

a query set modeled after distinct topics from forum posts from the AskDocs section of Reddit, designed to resemble laypeople's health queries. The same approach was followed by *Soldaini & Goharian (2017)*. Similarly, *Liu, Fang & Cai (2015)* selected question-like queries from topics of medical forums such as drugs.com, while *Zhang (2012)* selected tasks from Yahoo! Answers (to search for in MedlinePlus). Finally, *Azizan, Bakar & Rahman (2019)* used content from online forums, blogs, social media, and Google Instant to create a query set related to agriculture.

A major disadvantage of query generation *via* online communities is that the queries and their popularity can only be modeled in the context of the respective online community but not beyond. Whether or how often the queries generated in this manner are searched for *via* search engines remains unclear.

## Queries provided by subjects

The approach of queries provided by subjects encompasses all studies in which the authors use a group of subjects who are asked to generate queries based on certain specifications.

To obtain data generated by real people, *Bilal & Ellis (2011)* identified $N = 130$ tasks in the literature from 1989 to 2011 that were assigned to children and/or self-selected by them. Then the way children queried was examined, and the words used built the foundation of the query set. During the 2018 U.S. midterm elections, *Trielli & Diakopoulos (2022)* analyzed whether search results differ for members of different ideological groups. As a basis for the search results to be analyzed later, queries were needed. For this purpose, the authors conducted online surveys in several states, asking the subjects what terms they would use when searching for information about a candidate. To build a test collection (UQV100), *Bailey et al. (2016)* used crowdsourcing and collected $N = 5,764$ unique queries from $N = 263$ workers.

Using this approach has limitations. First, researchers receive queries from real people but not from a natural situation (using a search engine). In addition, self-reported behavior ("What search terms would you use?") does not necessarily reflect natural user behavior. Thus, it remains unclear whether the subjects would have searched in the way that they stated in the survey.

## Queries developed by the study authors

This approach includes all studies in which researchers develop the queries themselves. The authors do not consider whether or how frequently real search engine users use the queries, and they do not use predetermined lists of terms, as described in 'Predetermined lists of terms'.

The only basis for self-creating the queries are criteria such as query complexity (*Singer, Norbisrath & Lewandowski, 2012*), query type (*e.g.*, *Schultheiß, Sünkler & Lewandowski, 2018*; *Schultheiß, & Lewandowski, 2021*), or other criteria, such as the number of content farm articles per query (*McCreadie et al., 2012*). Queries developed by researchers is the least appropriate among the approaches described in this article since queries are arbitrarily arranged, and popularity is not considered. However, queries developed by authors can serve as a starting point (*i.e.*, the initial list of terms, see 'Selecting an initial list of terms') for creating further queries.

## Predetermined lists of terms

Another approach to generating search queries is using predetermined lists with different thematic focuses. The lists differ in terms of their coverage range from relatively small samples to complete lists, for example, of all political parties or the names of all candidates running for an election.

An exemplary study using a complete list is the analysis by *Hinz, Sünkler & Lewandowski (2023)*. For the 2021 German federal election, the authors analyzed whether candidates use search engine optimization (SEO) on their personal websites. The analysis was based on the complete list of all candidates in the election ($N = 6,211$). Other studies also used predetermined but sampled lists. For instance, *Torres & Rogers (2020)* combined the names of political parties with specific issues associated with the political agendas found on official party websites or in Facebook comments. *Hussain et al. (2019)* used keyword captions of images to form queries for a retrieval effectiveness study regarding image search engines. *Leontiadis, Moore & Christin (2011)* generated a query set focusing on search-redirection attacks. The authors issued a seed query ("no prescription Vicodin") and then collected search phrases found on the retrieved pages linking to websites the attackers wished to promote, for example, online pharmacies. Another approach using predetermined lists is the project "data donation" ("Datenspende"). A plugin installed in the browser of the participants ("donors") conducted searches for predefined terms at regular intervals and sent the results of the first search engine result page (SERP) back to the researchers (*Krafft, Gamer & Zweig, 2019*). However, the queries selected by the researchers were the precise names of political parties and selected politicians. Whether or how frequently search engine users actually used these queries remains unknown (*e.g.*, one can easily see from tools like Google Trends that the query "Bündnis90/Die Grünen" for the German Green party is searched only seldom, as the party is usually referred to as "Grüne" or "Die Grünen"). Another data donation study, with the same query selection limitations, focused on health-related queries (disease + clinical term; *Reber et al., 2020*).

To summarize, for queries on predetermined lists, it remains unknown whether and how frequently they were used by search engine users. While using lists of predetermined terms alone has limitations, lists can serve as a basis for the query set to be created by using the approach described in the following section.

## QUERY SAMPLER: AN APPROACH FOR GENERATING QUERY SETS

We propose an approach for generating a query set under the precondition that researchers do not have direct access to queries from a commercial search engine provider like Google but still aim for query sets that are representative in terms of query popularity and topic coverage. The approach aims to cover the search interest related to an initial list of terms by building extensive sets of queries on any topics. In doing so, we combine initial lists of terms and popularity data by utilizing keyword research tools.

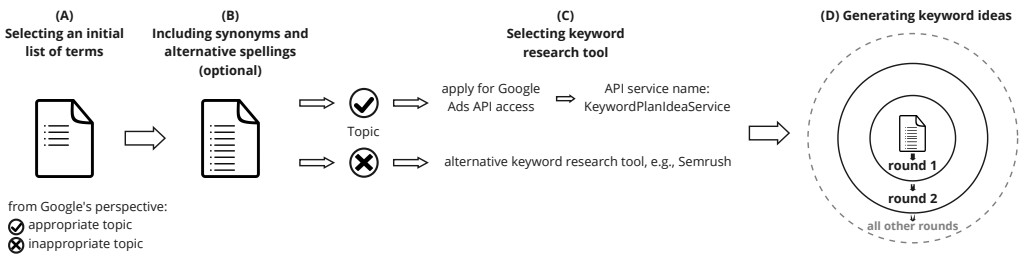

**Figure 1  Query sampler: an approach for generating query sets.** (A) Selecting an initial list of terms, (B) including synonyms and alternative spellings (optional), (C) selecting a keyword research tool, and (D) generating keyword ideas.

The approach is illustrated in Fig. 1 and consists of the following steps: (Fig. 1A) selecting an initial list of terms, (Fig. 1B) including synonyms and alternative spellings (optional), (Fig. 1C) selecting a keyword research tool, and (Fig. 1D) generating keyword ideas.

## Selecting an initial list of terms

First, a list of initial terms is selected. The initial list includes seed terms that form the basis of the query set to be created (*i.e.,* initial list = seed list for round 1). The entries of the initial list can come from various sources; for example, they can be compiled through brainstorming by the researchers or be predetermined lists with or without a thematic focus (see *Literature review*). Existing sources suitable for reuse are, for example, randomly selected Wikipedia articles (https://en.wikipedia.org/wiki/Special:Random) when a cross-topic list is to be created or Google Trends (https://osf.io/q7wt3) or Twitter Trends (https://twitter.com/i/trends) if trending topics are to form the basis of the list. Topic-specific lists, in contrast, can also come from multiple sources such as online communities, for example, the AskDocs section of Reddit (*Zuccon et al., 2016*), or public authorities, for example, a list of important terms regarding topics of domestic policy published by the German Federal Ministry of the Interior and Community (https://www.bmi.bund.de/DE/service/lexikon/lexikon-node.html) or a term index provided by the German Federal Department for Media Harmful to Young Persons (https://www.bzkj.de/resource/blob/197826/5e88ec66e545bcb196b7bf81fc6dd9e3/2-auflage-gefaehrdungsatlas-data.pdf).

## Including synonyms and alternative spellings (optional)

When the aim is to achieve high thematic coverage in the resulting query set, researchers should consider including synonyms and alternative spellings in the initial list of terms. It can be assumed that the approach described in this article will suggest several synonyms and alternative spellings, making this step obsolete, at least in theory. However, especially in the case of specialized vocabularies, such as medical terms, it cannot be assumed that this applies to all synonyms and alternative spellings. We illustrate the inclusion of synonyms and alternative spellings with a brief example in 'Synonyms and alternative spellings'.

## Selecting a keyword research tool

The next step is selecting a suitable keyword research tool (Fig. 1, step C). We explain the reasons for using Google services as the first choice from our point of view and the limitations of Google Keyword Planner and alternative tools.

We used Google Keyword Planning services for our approach for two main reasons. First, Google is the most popular search engine on the web. In the U.S., about 87% of all queries are submitted to Google (*StatCounter, 2023a*), and in Europe, 92% (*StatCounter, 2023b*). Therefore, Google achieves the highest coverage of the Internet user community, allowing more reliable statements on socially relevant topics. Second, even if it remains unclear how the keyword ideas are generated, their origin can be limited to Google, which does not apply to alternative tools as described at the end of this section.

However, using Google Keyword Planner (https://support.google.com/google-ads/answer/7337243?hl=en) through the standard user interface has limitations. For instance, the tool provides keyword ideas for up to 10 seed terms only (https://support.google.com/google-ads/answer/9327909?hl=en), which restricts its usefulness, especially when it comes to extensive lists of seed terms. Additionally, one must run an ad campaign of considerable size to obtain precise data on the average search volume of the generated keyword ideas. Otherwise, the Google Keyword Planner delivers only approximate search volume estimates (*e.g.*, 100–1,000; 10,000–100,000) (https://www.seroundtable.com/google-keyword-planner-throttled-22535.html), which are not very useful for research studies.

The limitations of the regular Google Keyword Planner do not apply to the Google Ads API. The Google Ads API enables users to generate large sets of keyword ideas, including precise data on search volume. To send requests to the Google Ads API, users need to authenticate the usage of their Google account *via* the Google Cloud Console. A client id (username) and client secret (password) are generated by creating a new project. Additionally, a refresh token must be generated by using the previous parameters. This token needs to be updated weekly to ensure the account security. The Python library GoogleAds requires these parameters together with the developer token and the ID of the Google Ads account to make calls to the API. Keyword ideas are generated by calling the KeywordPlanIdeaService (https://developers.google.com/google-ads/api/docs/keyword-planning/generate-keyword-ideas) and using the GenerateKeywordIdeasRequest. The input parameters are a keyword, location ID, and language id. No active ad campaign is necessary to use the API, as with the regular Google Keyword Planner, but a basic access token must be applied for (https://developers.google.com/adwords/api/docs/access-levels?hl=en). The application must include several details, such as the reasons for applying to use the API. Our statement that we will use the API for research led to the approval of our application.

Besides Google Keyword Planning services, a number of alternative tools are available, with Keyword Magic Tool from Semrush (https://de.semrush.com/analytics/keywordmagic/?q=adhs{&}db=de)[3] being among the most popular. Alternative keyword research tools are used when the query set to be created addresses a topic that is regarded as inappropriate by Google. Google does not serve ads for inappropriate content, which means that Google Keyword Planner does not provide any keyword ideas for such content either.

[3]Please note that Semrush is only meant to allow a comparison to Google services and is representative of many similar tools.

Google defines inappropriate content, among other things, as dangerous or derogatory content (*e.g.*, content promoting hate groups or hate group paraphernalia), sensitive events (*e.g.*, ads appearing to profit from a tragic event with no discernible benefit to users), or sexually explicit content (https://support.google.com/adspolicy/answer/6015406). Tools such as Semrush do not have such restrictions, so keyword ideas are generated even for content that Google considers inappropriate. One severe disadvantage of Semrush and similar tools is that the origin of the keyword ideas is not transparent. According to Semrush, the keyword ideas are based on data from third-party suppliers (https://www.semrush.com/kb/998-where-does-semrush-data-come-from). However, it remains unclear who the third-party suppliers are and which keyword idea comes from which source.

## Generating keyword ideas

We intend to cover the search interest related to the initial list of terms using the Google Ads API service "KeywordPlanIdeaService". Our approach is to resend the keyword ideas generated by the initial list of terms to the Google Ads API to gradually receive not only more but also more specific keyword ideas. As illustrated in Fig. 1, step D, this procedure is repeated in several rounds until no new keyword ideas emerge and saturation for the initial terms can be assumed (*Strauss & Corbin, 1998*, p. 143).

The process for each round is outlined below.

Round 1:

1. Collecting keyword ideas for all terms from the initial list
2. Cleaning the keyword ideas from ideas without search volume to ensure the criterion of popularity

Round 2 and all further rounds:

1. Collecting keyword ideas for all remaining keyword ideas from the previous round
2. Cleaning the keyword ideas from ideas without search volume to ensure the criterion of popularity
3. Removing duplicates within the same round (*i.e.,* keyword ideas generated by more than one initial term of the respective round)
4. Removing duplicates with previous rounds (*i.e.,* keyword ideas that have already been generated in a previous round)

## Proof of concept

To test our approach, we selected two published studies for comparison purposes. Study one is about the quality of information on cancer diet (*Herth et al., 2016*), and study two is on search engine optimization (SEO) for COVID-19 and radical right topics (*Lewandowski, Sünkler & Yagci, 2021*). The studies were selected for their differences in terms of topic and scope, allowing a first impression of the generalizability and scalability of our approach.

1. Both studies would have benefited from our approach, as a greater variety of queries and, thus, web pages would have strengthened the analyses.
2. The studies come from different subject areas. The radical right topics (*Lewandowski, Sünkler & Yagci, 2021*) allow a test of the described problem regarding Google's position

on inappropriate content (see 'Selecting a keyword research tool'), that is, whether keyword ideas are generated at all for such terms.

3. The initial lists of terms used in the studies vary in size.

We tested our approach by generating keyword ideas for the queries used in the studies and comparing the resulting keyword ideas in terms of number and search volume with the original studies. For better comparability with the example studies, we omitted considering synonyms and alternative spellings (see 'Including synonyms and alternative spellings (optional)').

## Study on cancer diet

[4] After consulting one of the authors, the use of the term "Krebsdiät" was confirmed since the original German term is not explicitly mentioned in the article.

The first study is from the medical field. The authors evaluate the quality of online patient information about cancer diet (*Herth et al., 2016*). For the term "Krebsdiät"[4] (English: "cancer diet"), the authors manually collected the first $N = 100$ organic results using the German version of Google and analyzed the quality of the results according to formal and content criteria, for example, transparency concerning provider and completeness. In their discussion, the authors present a short keyword analysis they conducted a few months after the study using Google Keyword Planner. The analysis showed that most users do not search for "cancer diet" but for more specific information on cancer diet or cancer diets by name, such as "ketogenic diet" or "nutrition in cancer". Hence, the authors conclude that a more detailed evaluation of patient information with more specified keywords is needed in future studies.

Table 2 shows keyword ideas we generated for the initial term "cancer diet" in five rounds. In columns two and three, the number of seed terms and the number of generated keyword ideas for these terms in each round are presented. In addition, the excluded keyword ideas are shown, that is, the number of keyword ideas with a search volume of 0, the number of duplicates within the current round, and the number of duplicates with already existing keyword ideas. The difference between generated and excluded keyword ideas is shown in the column of remaining keyword ideas. These form the basis ("seed terms") for the next round. The two rightmost columns show the search volume of the remaining keyword ideas per month, on average and sum.[5]

[5] Google Keyword Planning services provide aggregated search volume data for keywords, *e.g.*, "MBA" and their close variants, *e.g.*, "masters of business administration". Keywords and their close variants are reported with identical search volumes. Thus, the sum of the search volume may be higher than the *actual* search volume. See https://www.searchenginewatch.com/2016/09/26/reliable-search-volume-data-a-glimmer-of-hope/

In five rounds of collecting keyword ideas, we generated $N = 98$ unique keyword ideas (the sum of the remaining keyword ideas of each round) for the initial term "cancer diet". The keyword ideas of rounds three and four contain the highest number of duplicates. Due to excluding duplicates, the number of remaining keywords decreased considerably with each iteration. While from round two, 64% of the generated keyword ideas serve as seed terms for the next round, in round three, it is only 3%, and in round four, only one term. In round five, no new keyword ideas were generated. Together with the decreasing added monthly search volume in each round, this finding indicates a saturation regarding the general topic of the study (cancer diet).

[6] In our replication, "cancer diet" had an average search volume of $N = 149$ monthly searches.

The original study examined one term ("cancer diet") with a search volume of $N = 260$ average monthly searches at the time the study was conducted, according to the analysis by the authors.[6] By applying our approach, we expanded the term to a list of $N = 98$ terms, with an added search volume of $N = 2,144$ monthly searches on average. Thus, the

**Table 2  Generating keyword ideas for study on cancer diet.**

| | Seed terms (N) | Keyword ideas (N) | Exclusion of keyword ideas | | | Keyword ideas: remaining N (%) | Search volume of remaining keyword ideas (mean) | Search volume of remaining keyword ideas (Sum) |
| | | | Search volume of 0 N (%) | Duplicates within round N (%) | Duplicates with already existing keyword ideas N (%) | | | |
|---|---|---|---|---|---|---|---|---|
| Round 1 | 1 | 10 | 0 | 0 | 0 | 10 (100%) | 41 | 412 |
| Round 2 | 10 | 105 | 0 | 28 (27%) | 10 (10%) | 67 (64%) | 20 | 1,368 |
| Round 3 | 67 | 741 | 0 | 645 (87%) | 76 (10%) | 20 (3%) | 18 | 358 |
| Round 4 | 20 | 335 | 0 | 275 (82%) | 59 (18%) | 1 (0.3%) | 6 | 6 |
| Round 5 | 1 | 1 | 0 | 0 | 1 | 0 | 0 | 0 |
| Sum | | | | | | 98 | | 2,144 |

query used in the original study only covers 1% of queries and 12% of the projected search volume generated through our approach. By using our approach, researchers would have achieved a better evidential value even if they had cut off the list due to limited resources to analyze data for all queries. For instance, a cut-off after round two still would have covered $N = 77$ queries and 83% of the total search volume. This shows that researchers to do necessarily need to use all queries generated using our approach and still can increase the evidential value of their studies.

## Study on search engine optimization for radical right and COVID-19 topics

The second study we tested our approach on is about SEO (*Lewandowski, Sünkler & Yagci, 2021*). Using SEO indicators such as the usage of a site title, page speed, or usage of HTTPS, the authors built a rule-based classifier to determine the probability of SEO on a web page. To test the classifier, three query sets from Google Trends, including one on radical right content and one on the topic of COVID-19 were used. Through screen scraping, Google search results for the queries were collected and then classified according to their SEO probability. The results show that a large fraction of web pages found on Google are optimized (*Lewandowski, Sünkler & Yagci, 2021*). To test our approach, we used a sample of $N = 15$ queries of the COVID-19 ($N = 271$) dataset[7] and the full dataset of the radical right ($N = 82$) queries.

As Table 3 shows, we generated $N = 385$ keyword ideas for the COVID-19-related queries in three rounds since no new keyword ideas were generated in round three. Most keyword ideas were delivered in round one. As in the previous study on cancer diet, many keyword ideas were excluded because they were duplicates. The queries used in the sample ($N = 15$) from the original study cover 4% of the queries we cover with our approach. For the initial terms of the published study, we identified an aggregated search volume of $N = 1,473,536$ monthly searches, which is 35% of the search volume generated by our approach ($N = 4,170,514$ monthly searches).

[7] As "dataset", we refer to the initial lists of terms included in the research data. In three columns, the research data lists (1) the keyword ideas, (2) the round in which the idea was generated, and (3) the search volume of the respective keyword idea. Keyword ideas that are indicated with round "0" are the terms of the initial lists that were used to generate the keyword ideas in all further rounds.

**Table 3  Generating keyword ideas for COVID-19 queries.**

| | Seed terms (N) | Keyword ideas (N) | Exclusion of keyword ideas | | | Keyword ideas: remaining N (%) | Search volume of remaining keyword ideas (mean) | Search volume of remaining keyword ideas (Sum) |
|---|---|---|---|---|---|---|---|---|
| | | | Search volume of 0 N (%) | Duplicates within round N (%) | Duplicates with already existing keyword ideas N (%) | | | |
| Round 1 | 15 | 307 | 5 (2%) | 0 | 5 (2%) | 297 (97%) | 4,775 | 1,413,482 |
| Round 2 | 297 | 2,970 | 79 (3%) | 2,205 (84%) | 301 (10%) | 88 (3%) | 31,330 | 2,757,032 |
| Round 3 | 88 | 35 | 24 (68%) | 3 (9%) | 8 (23%) | 0 | | |
| Sum | | | | | | 385 | | 4,170,514 |

**Table 4  Generating keyword ideas for radical right queries.**

| | Seed terms (N) | Keyword ideas (N) | Exclusion of keyword ideas | | | Keyword ideas: remaining N (%) | Search volume of remaining keyword ideas (mean) | Search volume of remaining keyword ideas (Sum) |
|---|---|---|---|---|---|---|---|---|
| | | | Search volume of 0 N (%) | Duplicates within round N (%) | Duplicates with already existing keyword ideas N (%) | | | |
| Round 1 | 82 | 233 | 37 (16%) | 2 (1%) | 32 (14%) | 162 (70%) | 993 | 160,819 |
| Round 2 | 162 | 854 | 26 (3%) | 575 (67%) | 165 (19%) | 88 (10%) | 51 | 4,486 |
| Round 3 | 88 | 1,194 | 24 (2%) | 1,020 (85%) | 127 (11%) | 23 (2%) | 33 | 760 |
| Round 4 | 23 | 71 | 0 | 30 (42%) | 36 (51%) | 5 (7%) | 32 | 162 |
| Round 5 | 5 | 0 | 0 | 0 | 0 | 0 | 0 | 0 |
| Sum | | | | | | 278 | | 166,227 |

For the radical right queries, we generated $N = 278$ keyword ideas in five rounds, as shown in Table 4. The queries used in the original study cover 29% of queries we cover with our approach. For the list of initial terms of the published study, we identified an aggregated search volume of $N = 27,117$ monthly searches, which is 16% of the search volume generated by our approach ($N = 166,227$ monthly searches). No ideas were generated for 46% of the initial terms ($N = 38$), including right-wing extremist numeric codes such as "1488". It can be assumed that Google classifies such unambiguous right-wing terms as inappropriate, so no keyword ideas are generated (see the explanation in 'Selecting a keyword research tool').

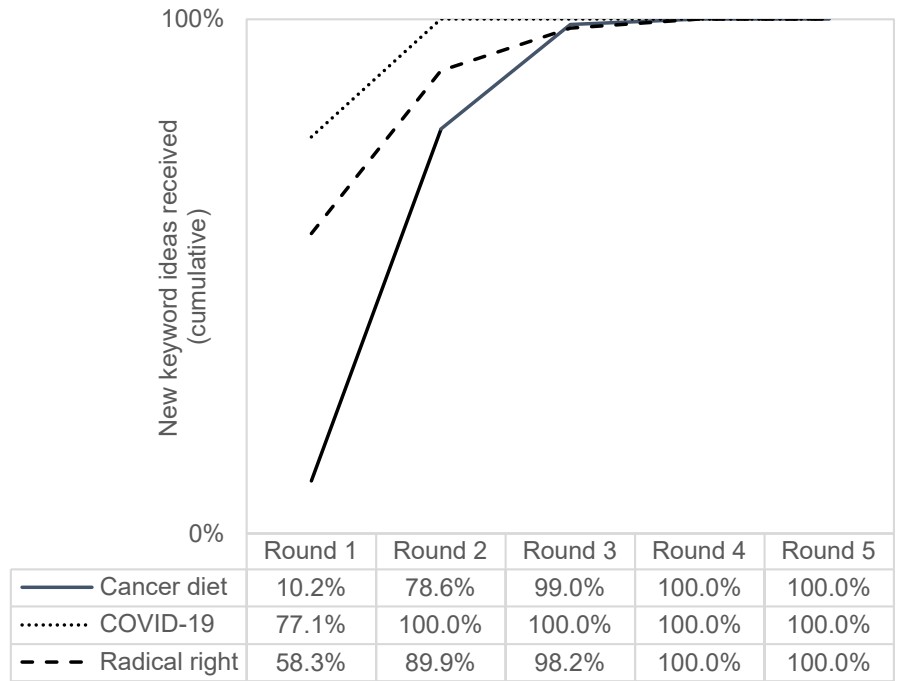

| | Round 1 | Round 2 | Round 3 | Round 4 | Round 5 |
|---|---|---|---|---|---|
| —— Cancer diet | 10.2% | 78.6% | 99.0% | 100.0% | 100.0% |
| ········· COVID-19 | 77.1% | 100.0% | 100.0% | 100.0% | 100.0% |
| – – – Radical right | 58.3% | 89.9% | 98.2% | 100.0% | 100.0% |

**Figure 2** **Keyword ideas of all studies and rounds (cumulative).**

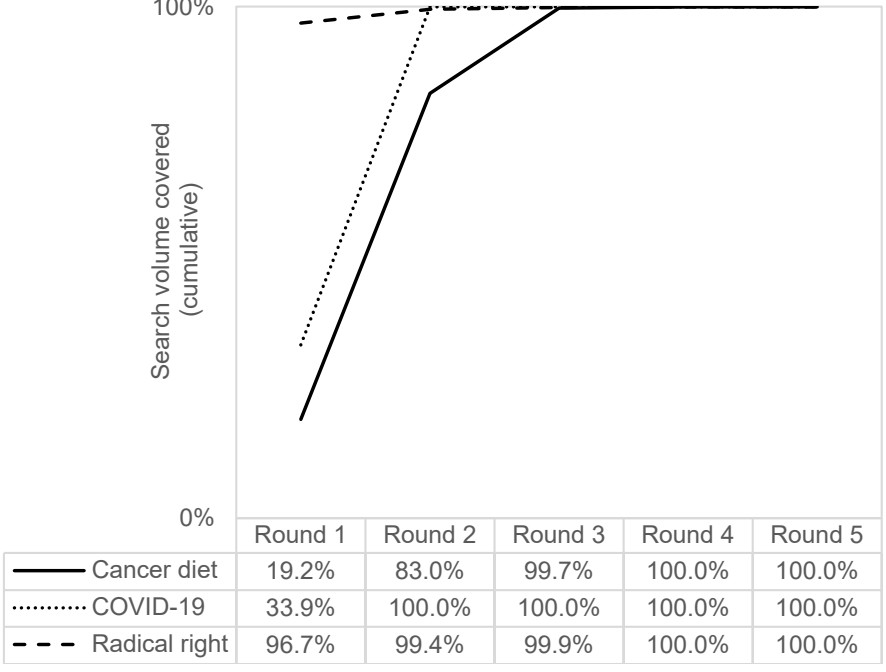

| | Round 1 | Round 2 | Round 3 | Round 4 | Round 5 |
|---|---|---|---|---|---|
| —— Cancer diet | 19.2% | 83.0% | 99.7% | 100.0% | 100.0% |
| ········· COVID-19 | 33.9% | 100.0% | 100.0% | 100.0% | 100.0% |
| – – – Radical right | 96.7% | 99.4% | 99.9% | 100.0% | 100.0% |

**Figure 3** **Search volume of all studies and rounds (cumulative).**

## Cumulative keyword ideas and search volume of all studies

The following figures show the cumulative keyword ideas (Fig. 2) and search volume (Fig. 3) of all studies and rounds, that is, the sum of keyword ideas and search volume that have been added up to a certain round.

From Fig. 2, we see that the most keyword ideas were collected after two rounds. It is noticeable that for "cancer diet" the highest gain in new keyword ideas was achieved from round one to round two. This can be explained by the fact that the initial list of terms consisted of only one query (namely "cancer diet"), while the lists of terms for the other two topics were more extensive ($N = 15$ queries for COVID-19 and $N = 82$ queries for radical right). No new ideas were generated after four rounds (at the latest).

From the cumulative search volume shown in Fig. 3, differences between the dataset on radical right topics and the other two datasets become clear. The search volume of the keyword ideas on cancer diet and COVID-19 increased considerably in round two (from 19.2% to 83% and from 33.9% to 100%, respectively). However, the situation is different with the radical right queries. A total of 96.7% of their search volume was already covered in round one, so the further rounds increased the search volume only slightly.

## Synonyms and alternative spellings

Here, we illustrate with a short example that it is worthwhile to include synonyms and alternative spellings in the initial list of terms to achieve high thematic coverage of the resulting query set (see 'Including synonyms and alternative spellings (optional)'). In her study on skin diseases, *Alby (2020)* built an initial list of queries related to spinalioma[8] together with $N = 15$ synonyms and alternative spellings. She then entered all initial terms into Google Keyword Planner and collected keyword ideas. For all keyword ideas, Alby collected Google results and analyzed them (*e.g.*, regarding their information quality). We repeated Alby's approach by using Google Ads API and generated $N = 1,616$ keyword ideas with only $N = 108$ (7%) duplicates for "spinalioma" together with the synonyms and alternative spellings. Thus, due to the low duplicate rate, it is worthwhile to include synonyms and alternative spellings in the initial list of terms since many new keyword ideas can be generated.

## DISCUSSION

This article describes an approach to sample queries from commercial search engines using keyword research tools. First, an initial list of terms is selected and keyword ideas are generated (round one of collecting keyword ideas) from this list. These keyword ideas then serve as seed terms, that is, the starting point, for the second round to collect more keyword ideas. This procedure is repeated in further rounds until no new keyword ideas are found and, therefore, the initial list is saturated. The seed terms used to generate keyword ideas in round one are the researcher's predefined terms, whereas from round two onwards, the seed terms are the keyword ideas received in the preceding round.

We empirically tested our approach by using the queries of two published studies as initial terms. Study one is about information quality regarding cancer diet (*Herth et al., 2016*), and study two is on SEO for COVID-19 and radical right topics (*Lewandowski,*

[8]The German term used in the study is "Spinaliom".

*Sünkler & Yagci, 2021*). The number of queries and the total search volume covered could be significantly expanded when comparing the original studies with our query collection approach. After three rounds of collecting keyword ideas, no more new ideas were generated. Two rounds were sufficient to cover most of the total generated keyword ideas and search volume. Hence, both studies would have benefited from our approach, as the foundation of the studies, that is, the search results analyzed, would have been more consistent with what users really search for and see on the web.

Our approach considers the popularity of the queries and allows to cover a self-selected topic comprehensively. Both are advantages over other approaches for generating query sets identified in the literature. Previous approaches either consider the popularity of the queries only to a limited extent, for example, by extracting content from online communities, or do not or only partially allow for full thematic coverage, for example, when using queries developed by researchers. Both prerequisites, considering query popularity and allowing topic coverage, are also met by queries delivered by search engine providers, which, however, are only made available to few researchers for specific purposes.

The described approach and its testing come with several limitations. Firstly, when choosing the keyword research tool, it should be noted that a dependency on a provider, such as Google, arises. This dependency is also reflected in the required application for API access. If access is not granted or withdrawn, the implementation of our approach in its current form is no longer possible. Second, the studies we used to test the approach illustrate only a fraction of the possible use cases. Third, a content analysis of the generated keyword ideas has yet to be performed.

These limitations point to the need for future studies. First, the approach should be conducted with other keyword research tools, and the results should be compared; this would make the results more reliable. Moreover, this could counteract the dependency on one provider. Second, the approach should be tested on other topics and with more extensive initial lists of terms to check the applicability and scalability beyond the replicated studies. Third, an analysis should be conducted to identify possible topic drifts for the generated keyword ideas, for example, through human evaluators. Topic drifts could occur if the topic of the initial terms is no longer reflected at a certain point, for example, after a particular round of generating keyword ideas (*Hobbs, 1990*). Topic drifts must be identified to exclude affected keyword ideas. Fourth, the effects of expanded query sets on study outcomes should be examined. This is particularly important for studies that aim to make statements about the quality of information a user is confronted with when searching, for example, for health-related topics. The study results may also change when the number of examined queries grows. Finally, it needs to be discussed how to deal with the so-called close variants, which are output by Google Keyword Planning services (see 'Study on cancer diet'). As close variants lead to an unrealistically high total search volume of the query set, they will likely have to be excluded.

The approach described in this article for generating a query set has many possible applications since it can be applied flexibly to different topics and is relatively straightforward to implement. Studies investigating search results related to social phenomena would particularly benefit from the approach. The search interest of real

users is covered by systematically obtaining keyword ideas for an initial list of terms. When researchers retrieve and analyze search results on this basis, the search results are more likely to correspond to those seen by real search engine users than if queries without user reference, such as queries developed by researchers or using only technical terms, had been examined. This is especially relevant when statements about social phenomena are to be made, for example, when examining the quality of patient information on the Internet. Otherwise, authors risk analyzing search results that users do not see with their queries, limiting the reliability of their study results.

## CONCLUSION

In this article, we described an approach to sample queries from commercial search engines using Google Keyword Planning services. We empirically tested our approach with two published studies on the quality of patient information and SEO. The results show that the number of queries and total search volume could be significantly expanded. When comparing the *number of queries* of the original studies with the queries generated by our approach, the original studies only cover 1% (cancer diet), 29% (radical right), and 4% (COVID-19) of the queries generated by our approach. The same holds for the *total search volume*. The search volume of the queries of the original studies only covers 12% (cancer diet), 16% (radical right), and 35% (COVID-19) of the search volume generated by our approach. This leads us to the conclusion that the studies would have benefited from our approach since the queries generated by our approach better reflect actual user behavior. In general, we found that researchers can improve the evidential value of studies that use search queries by extending their initial query set by using our approach. Thus, the approach offers a wide range of applications for studies that seek to draw conclusions about social phenomena using search engine queries. The approach is relatively easy to apply to different topics and use cases. Future research should test the approach with other keyword research tools and topics and conduct content analyses of the generated keyword ideas.

### Funding

This work is funded by the German Research Foundation (DFG Deutsche Forschungs-gemeinschaft), grant number 467027676. We received support for the article processing charge from the Open Access Publication Fund of Hamburg University of Applied Sciences. The funders had no role in study design, data collection and analysis, decision to publish, or preparation of the manuscript.

### Grant Disclosures

The following grant information was disclosed by the authors:
The German Research Foundation (DFG Deutsche Forschungsgemeinschaft): 467027676.
Open Access Publication Fund of Hamburg University of Applied Sciences.

## Competing Interests

The authors declare there are no competing interests.

## Author Contributions

- Sebastian Schultheiß conceived and designed the experiments, performed the experiments, analyzed the data, prepared figures and/or tables, authored or reviewed drafts of the article, and approved the final draft.
- Dirk Lewandowski conceived and designed the experiments, performed the experiments, authored or reviewed drafts of the article, and approved the final draft.
- Sonja von Mach conceived and designed the experiments, performed the experiments, authored or reviewed drafts of the article, and approved the final draft.
- Nurce Yagci performed the computation work, authored or reviewed drafts of the article, and approved the final draft.

## Data Availability

The query sets from the proof of concept are available at OSF: Schultheiß, Sebastian, Dirk Lewandowski, Sonja von Mach, and Nurce Yagci. 2023. "Query Set." OSF. January 16. https://doi.org/10.17605/OSF.IO/S65JD.

The code for querying Google Ads API is available at Zenodo: Nurce Yagci. (2023). yagci/keyword-planner: source code (Version v2). Zenodo. https://doi.org/10.5281/zenodo.7828931.

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
