# Peer review of "Query sampler: generating query sets for analyzing search engines using keyword research tools"

_PeerJ Computer Science, doi:10.7717/peerj-cs.1421_

## Round 0.1 · original submission · Minor Revisions

The reviewers recognized the contributions of the work. However, some issues need to be solved before further consideration.

Reviewer 1 ·

Basic reporting

The study: “ It all starts with the query: Generating query sets for analyzing search engines using keyword research tools” described an approach to sample queries from commercial search engines, using keyword research tools, it’s Clear and unambiguous, and professional English is used, the Literature references and sufficient field background/context are provided.
however my concern :
- The title is so long and needs to be more informative and relevant.
- The aim of the study is clear however the abstract needs to explain how the results can be effectively implemented.
- The Abstract doesn’t explain clearly the foundation of the study .
- Figures 2 and 3 show “Cumulative keyword ideas and the search volume of all studies need more explanation by addressing more details in the section: ” Cumulative keyword ideas and search volume of all studies.”.
- The research gap needs to be mentioned in subsection.
- The paper is shown in a professional way however it’s suggested to list research questions.
- It’s suggested to add the study contributions.
- The results include clear definitions of all terms however it’s suggested to describe in brief the features of the datasets used.

Experimental design

The approach is defined and measured appropriately, the methods valid and reliable and there is enough detail mentioned. However, it’s suggested to add a table summarizing and comparing the results with similar approaches.
However, the research fills an identified knowledge gap but it’s suggested to add research questions.

Validity of the findings

All underlying data have been provided; they are robust, statistically, however, the conclusion needs to be supported by references or results

·

Basic reporting

The authors presents a clear overview of the problem and provides sufficient detail on the approach and its evaluation. The writing is clear and easy to follow, and the authors provide a thoughtful analysis of the implications and limitations of their approach. The article is well-suited for publication

Experimental design

The experimental design is well-structured and clearly presented in the article. The author provides sufficient detail on the selection of the studies.

Validity of the findings

The author's expertise in sampling search engine queries for studies greatly contributes to the validity of the research.

Additional comments

Overall, the paper addresses an important problem in the field of social research and presents a novel approach to address it.

The introduction provides a clear overview of the problem and its significance,

The methodology section is well-written and provides sufficient detail to replicate the study.

The empirical evaluation section provides a clear description of the results and the comparison with the original studies.

The discussion section provides a thoughtful analysis of the implications of the approach and its limitations.

The paper is generally well-written and easy to follow,

Overall, the paper presents a novel and useful approach to sampling search engine queries for social research. The authors have provided sufficient detail to allow for replication, and the empirical evaluation shows promising results.

---

## Round 0.2 · accepted · Accept

Congrats to the authors and thanks for the efforts to improve the article.